# Stress-Induced Magnetic Anisotropy in Fe-Based Amorphous/Nanocrystalline Alloys: Mechanisms, Advances and Challenges

**DOI:** 10.3390/ma18071499

**Published:** 2025-03-27

**Authors:** Jianqiang Zhang, Yanjun Qin, Xiaobin Liu, Yuxiang Zhao, Wenqiang Dang, Xiaozhen Fan, Xinyi Chen, Yuanrong Yu, Zixuan Yang, Shipeng Gao, Duanqiang Wu, Yunzhang Fang

**Affiliations:** 1College of Electronic Information and Electrical Engineering, Tianshui Normal University, Tianshui 741001, China; liuxb@tsnu.edu.cn (X.L.); zhaoyx@tsnu.edu.cn (Y.Z.); dangwenqiang@tsnu.edu.cn (W.D.); cxy200456xyz@163.com (X.C.); yyr61740472198@163.com (Y.Y.); yzx20051202@163.com (Z.Y.); gsp0223@163.com (S.G.); awdq1122@163.com (D.W.); 2School of Physics and Electronic Information Engineering, Zhejiang Normal University, Jinhua 321004, China; fxz@zjnu.cn (X.F.); fyz@zjnu.cn (Y.F.)

**Keywords:** stress-induced magnetic anisotropy, magneto-elastic coupling, lattice plane anisotropy, anisotropic distribution of nanograins, in situ technique

## Abstract

Fe-based amorphous and nanocrystalline alloys, such as FINEMET and its improved variants, are highly valued as green energy-saving materials due to their unique magnetic properties, including high permeability, low coercivity, and near-zero saturation magnetostriction. These characteristics have enabled their extensive use in power electronics and information technology. However, the full potential of these alloys remains unfulfilled due to insufficient understanding of their stress sensitivity. This study focuses on the development history, heat treatment, annealing processes, chemical composition, and underlying mechanisms of Fe-based amorphous and nanocrystalline alloys, aiming to provide insights into stress-induced magnetic anisotropy and guide the development of greener and more efficient soft magnetic materials.

## 1. Introduction

Metals have always been essential in shaping human civilization, with each innovation in metal materials driving significant progress in technology and industry. A major breakthrough occurred in the 1960s when Duwez et al. [1,2] introduced rapid solidification techniques, leading to the first Fe-based amorphous soft magnetic alloy (Fe_80_P_12.5_C_7.5_). This innovation expanded possibilities for optimizing composition and heat treatment processes. Building on this, Yoshizawa et al. [3] at Hitachi Metals developed a dual-phase structure with α-Fe(Si) nanocrystals in an amorphous matrix, significantly enhancing magnetic properties [4,5]. This study inspired further development of high-saturation induction (Bs) Fe-based nanocrystalline alloys like NANOPERM [6], HITPERM [7], and NANOMET [8,9], which adopted similar principles and optimized heat treatment processes.

To meet modern industrial demands, researchers proposed stress annealing (SA) to tune magnetic properties by inducing magnetic anisotropy (MA) during heat treatment [10]. Studies have shown that applying stress during nanocrystallization efficiently induces MA through the magneto-elastic coupling (MEC) effect, which links MA to residual stress and magnetostriction [11,12,13]. At the microscopic level, Ohnuma et al. [14] observed lattice plane anisotropy (LPA) in α-Fe(Si) nanocrystals under stress, providing evidence for stress-induced MA. Fang et al. [15,16,17] further revealed that stress induces significant LPA, unlike free annealing conditions. However, several critical questions remain unresolved, such as the dynamic correlation between microstructure and annealing stress and the reversibility of MA. These questions lead to an unfilled understanding of the stress-sensibility of Fe-based alloys.

This study aims to provide a comprehensive overview of the development of Fe-based amorphous and nanocrystalline soft magnetic alloys, focusing on heat treatment methods, annealing parameters, chemical composition, and underlying mechanisms for inducing MA. By synthesizing these aspects, we hope to offer a roadmap for further developing high-performance materials and optimizing them for green and efficient technological applications.

## 2. Development History of Fe-Based Soft Magnetic Alloys

Fe-based soft magnetic alloys are indispensable in modern technology, industry, and defense, developing alongside the progress in electricity, electronics, and magnetic fundamental theories. Since the advent of Fe-based amorphous soft magnetic alloys, their development can be summarized into three stages [18], as shown in Figure 1.

### 2.1. Development and Application of Fe-Based Amorphous Soft Magnetic Alloys

Initially, Gubanov theoretically predicted the strong macroscopic magnetism of amorphous alloys, later validated by Duwez’s Fe_80_P_12.5_C_7.5_ alloy [1]. In 1969, Pond and Maddin [19] invented the melt-spinning technique for alloy melts and produced amorphous alloy ribbons tens of meters long, enabling continuous production of amorphous ribbons. Subsequent developments included Fe-P-C, Fe-P-B, Fe(Co,Ni)-Si-B, and Fe(Co,Ni)-(Zr,Nb,Hf)-B systems. Compared to crystalline alloys, Fe-based amorphous alloys exhibit low hysteresis and eddy current losses due to near-zero magneto-crystalline anisotropy and high resistivity. In 1979, Allied Signal (with Narasimhan as the main inventor) developed the planar flow casting technology [20], which provided a technical guarantee for the commercial application of Fe-based amorphous soft magnetic alloys. Based on this technology, the METGLAS series of Fe-based amorphous soft magnetic alloys were successively produced, steering the research of Fe-based amorphous soft magnetic alloys towards industrialization and commercialization. These alloys gradually found applications in power electronics, such as transformers, reactors, and electromagnetic shielding devices. In 1984, a U.S. transformer manufacturer showcased an amorphous alloy distribution transformer at the IEEE conference. During the same period, Japan and Germany also made progress in the application and development of amorphous soft magnetic alloys, primarily in the area of power electronic components, thus propelling the research and application of Fe-based amorphous soft magnetic alloys to a climax [18].

### 2.2. Development and Application of Fe-Based Nanocrystalline Soft Magnetic Alloys

In 1988, Yoshizawa et al. [3] introduced appropriate amounts of Cu and Nb into the FeSiB alloy system, creating FINEMET alloy via amorphous crystallization. The typical composition is Fe_73.5_Cu_1_Nb_3_Si_13.5_B_9_, with the production process illustrated in Figure 2 [21]. The unique dual-phase structure endowed FINEMET with high permeability, low coercivity, near-zero magnetostriction, and reduced magneto-crystalline anisotropy (MCA). As shown in Figure 3, although the saturation magnetic induction strength (Bs = 1.24 T) of the FINEMET alloy is lower than that of commonly used amorphous alloys and silicon steel, FINEMET combined the high Bs of Fe-based amorphous alloys with the high permeability of Co-based alloys, sparking renewed interest in Fe-based systems [18].

Despite its excellent comprehensive soft magnetic properties, the relatively low Bs of FINEMET alloy remains a challenge to overcome in the context of the miniaturization trend of power electronic devices. It is well known that the Bs of Fe-based soft magnetic alloys is generally proportional to the content of Fe. To address the demand for higher Bs, Suzuki et al. [22] developed the Fe_91_Zr_7_B_2_ nanocrystalline alloy system (known as NANOPERM alloy) in 1991, demonstrating advantages in Bs, but it lagged behind the FINEMET alloy in terms of coercivity and effective permeability. Additionally, its low amorphous-forming ability made it difficult to commercialize at a low cost. Later, in 1998, Willard et al. [7] developed the FeCoMBCu (M = Zr, Nb, Hf) nanocrystalline alloy (known as HITPERM alloy), exhibiting higher Bs and enhanced thermal stability. But its costs increased due to Co addition., making it unf avorable for commercial application. In 2006, Ogawa et al. [23] developed Fe-based amorphous alloys with compositions of Fe_81.7_Si_2_B_16_C_0.3_ and Fe_82_Si_2_B_14_C_2_ with Bs values of up to 1.64 T and 1.67 T, respectively. Research also found that the losses under operating conditions of 50 Hz and 1.4 T were reduced by 15%. This finding attracted widespread attention from the scientific and industrial communities. Although there are still many challenges in industrialization, this study indicates a promising direction for further improving the Bs of Fe-based amorphous soft magnetic alloys without the addition of precious metals [24,25].

**Figure 3 materials-18-01499-f003:**
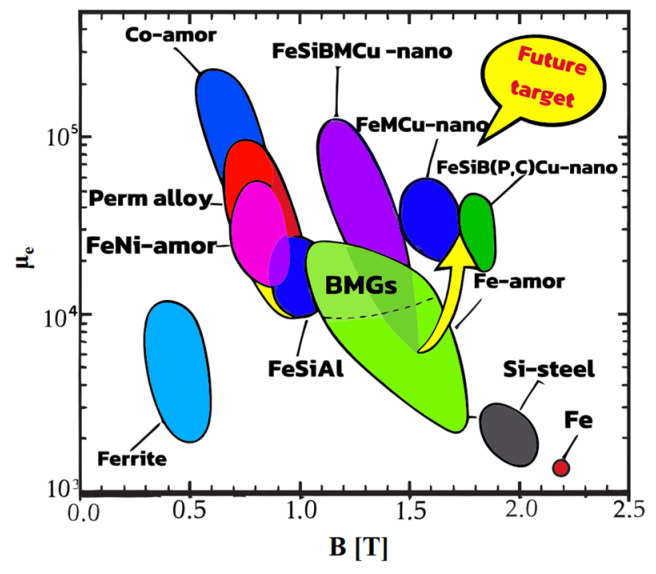
Saturation magnetic induction and effective permeability (1 kHz) of typical soft magnetic alloy materials (Li, F.C. 2019 [25]).

### 2.3. Development and Application of Fe-Based Soft Magnetic Alloys with High Bs

In 2009, Makino et al. [8,9] developed the Fe-Si-B-P-Cu nanocrystalline alloy system, known as the NANOMET alloy, which exhibited a Bs of 1.9 T. This alloy rivaled silicon steel in terms of Bs while maintaining high permeability and low coercivity (Hc), thus attracting widespread attention. However, the alloy system had relatively low amorphous-forming ability, and its nanocrystallization process was stringent, making it difficult to commercialize. Despite these challenges, its development sparked the third wave of research enthusiasm for Fe-based soft magnetic alloys.

During this wave of research, scholars in China achieved many important results [26,27,28,29]. For example, Fan et al. [26] reported an excellent soft magnetic Fe_83_B_10_C_6_Cu_3_ nanocrystalline alloy with Bs = 1.78 T, Hc = 5 A/m, and core loss P1/5 = 0.34 W/kg. They found that the Hc of the Fe_84−x_B_10_C_6_Cu_x_ alloy decreased with increasing Cu content and reached a minimum value in the alloy with x = 1. Li et al. [28] proposed a new nanostructural design concept, reporting an Fe_85.5_Si_10_B_2_P_2_C_0.5_ amorphous nanocrystalline soft magnetic alloy with a Bs as high as 1.87 T and effective permeability of 1~2.5 × 10^4^. This was achieved by densifying pre-nucleation sites and refining nanostructures through transient metal-rich interfaces during melt quenching with critical cooling rates. This heterogeneous structure and lean alloying design concept provided a model for the development of next-generation high Bs soft magnetic alloy materials. Shen et al. [29] designed a new type of Fe_84−x−z_Si_4−y_B_8.4_P_3.6_Ni_x_Mo_y_Cu_z_ nanocrystalline soft magnetic alloy using the co-alloying effect of Ni, Mo, and Cu. They found that the addition of a small amount of Ni (<1.5%) could reduce Hc without sacrificing Bs, while co-alloying refined the nanocrystal size from 28.5 nm to 13.1 nm and exhibited low Hc (1~2 A/m) and high Bs (1.81 T) characteristics.

After more than half a century of development, Fe-based amorphous and nanocrystalline soft magnetic alloys with excellent performance have emerged, as shown in Table 1 [18]. These alloys are made into magnetic cores to meet different application needs in the fields of power and electronics, such as sensors, magnetic amplifiers, high-frequency transformers, inductor cores, common-mode chokes, distribution transformers, photovoltaic inverters, wind power converters, and onboard chargers [25] as showed in Figure 4. However, in the face of increasingly severe environmental pollution and energy crises, there is an urgent need to develop new types of soft magnetic alloy materials that are efficient, energy-saving, and environmentally friendly. Innovating alloy design concepts and optimizing heat treatment processes are important measures to achieve this goal. For the original FINEMET alloy, the optimal heat treatment temperature is typically around 550 °C. However, for its improved variants, such as Co-FINEMET and Cr-FINEMET, the optimal heat treatment temperature may need to be adjusted to the range of 520–580 °C, depending on the specific doping elements and their concentrations.

## 3. Annealing Methods for Inducing Magnetic Anisotropy

The most effective approach for inducing MA in Fe-based alloys is through the coupling of stress and temperature fields during their crystallization process. This method promotes the formation of a nanobiphasic structure, which is crucial for enhancing the magnetic properties of these materials. The crystallization of Fe-based alloys generally proceeds through two distinct stages, as illustrated in Equation (1):
(1)Amorphous→α+amorphous→α+β,

In this context, α denotes the primary crystalline phase that precipitates from the amorphous matrix, such as the α-Fe phase, while β represents intermetallic compound phases, such as Fe_2_B or Fe_3_B. Figure 5 presents the differential scanning calorimetry (DSC) curves of Fe_81.7−y_Si_4_B_13_Cu_1.3_Nb_y_ alloys [30], with the two exothermic peaks corresponding to the stages [31] described in Equation (1). To achieve a biphasic heterogeneous structure composed of the primary crystalline phase and the amorphous matrix while avoiding the formation of intermetallic compounds in the second stage, the crystallization temperature should be carefully selected to fall between the first and second crystallization temperatures (ΔTx = Tx_2_ − Tx_1_). For the optimized composition of the FINEMET alloy, the literature consistently reports that the optimal crystallization temperature is around 550 °C.

### 3.1. Conventional Stress Annealing

Conventional SA is a widely employed heat treatment technique for inducing MA in ferromagnetic materials by coupling temperature and stress fields [32,33,34,35,36]. This method involves placing the material inside a tubular furnace under a protective gas or vacuum environment. One end of the material is securely fixed by a fixture, while the other end is subjected to a controlled mechanical load along the length of the furnace (typically 1–2 m in length). The furnace used in conventional stress annealing has an inner diameter of approximately 0.1–0.2 m, and the working zone length is around 0.5–1 m. These dimensions ensure uniform heating and effective stress application during the annealing process. The annealing process consists of three distinct stages:

Heating Stage: The material is heated from room temperature to the desired annealing temperature at a controlled rate, typically in the range of 520–580 °C. This temperature range is chosen to ensure the formation of a stable nanobiphasic structure while avoiding excessive grain growth or the formation of undesirable intermetallic phases. This stage primarily involves atomic relaxation of the amorphous precursor and nucleation of the primary crystalline phase, such as α-Fe. Isothermal Holding Stage: The material is held at the annealing temperature for a specified duration. During this stage, the primary crystalline phase grows rapidly, forming a nanoscale biphasic structure composed of the α-Fe and the residual amorphous matrix. This process is facilitated by the diffusion inhibition mechanism of large metal atoms, which helps refine the microstructure [32,33,34,35,36]. Cooling Stage: The material is then cooled to room temperature along with the furnace. This step effectively “freezes” the nanobiphasic structure formed during the isothermal holding stage into a relatively stable energy state [32,33,34,35,36].

Figure 6 presents a schematic diagram of a conventional SA device. The entire annealing process is closely monitored by a computer controller, which allows real-time detection of annealing temperature, applied stress, and annealing time. Additionally, the controller can measure the tensile strain of the material during annealing. This precise control ensures consistent and reproducible results. Conventional SA is an effective method for inducing uniform uniaxial MA in ferromagnetic materials. The magnitude of the induced MA is highly dependent on annealing parameters, including temperature, time, stress, and heating and cooling rates [32,33,34,35,36]. Studies have shown that the MA induced by conventional SA is two orders of magnitude higher than that induced by an external magnetic field [14,17,37]. This significant enhancement in MA underscores the importance of optimizing annealing parameters to achieve desired magnetic properties in Fe-based nanocrystalline alloys.

### 3.2. Rapid Stress Annealing

Rapid SA has emerged as a significant advancement in material heat treatment technologies in the early 21st century, aiming to efficiently regulate and optimize the magnetic properties of materials [38,39,40,41,42,43]. This technique is particularly effective for high-Bs Fe-based nanocrystalline alloys, where the goal is to reduce coercivity while maintaining excellent soft magnetic properties [44,45,46,47,48,49,50,51]. Figure 7 illustrates two types of rapid SA devices reported in the literature [45,51]. In this method, the material is subjected to tensile stress and continuously passes through a uniform heating zone (or heating roller) to ensure effective heat transfer. The annealing temperature for rapid SA is typically at least 100 °C higher than that of conventional SA, with annealing times ranging from a few seconds to tens of seconds. This technique features significantly higher heating and cooling rates, which have been shown to be effective in reducing the coercivity of Fe-based nanocrystalline alloys [45,46].

The thermal history of the amorphous alloy plays a crucial role in its crystallization behavior. By altering the thermal history, the crystallization process can shift between homogeneous and heterogeneous nucleation. High heating rates provide a broader crystallization temperature window for amorphous alloys and can change the relaxation behavior of the pre-crystallization structure. Recent studies on high-Bs Fe-based nanocrystalline alloys have demonstrated that a high heating rate is essential for maintaining excellent soft magnetic properties after nanocrystallization. For example, the NANOMET alloy requires a heating rate of 10 K/s to achieve optimal properties. Additionally, studies on Fe_86–87_B_13–14_ nanocrystalline alloys with Bs above 1.9 T have shown that increasing the heating rate from 1.7 K/s to nearly 10^4^ K/s can reduce the grain size from 40 nm to 20 nm and the coercivity from 150 A/m to 6–7 A/m [49,50]. These findings highlight the importance of rapid SA in optimizing the magnetic properties of Fe-based alloys.

### 3.3. Stress Annealing Combined with Magnetic Field

SA combined with a magnetic field has emerged as a powerful technique for enhancing the magnetic properties of soft magnetic materials. This method, known as stress magnetic field annealing, involves subjecting the material to both stress and a magnetic field (provided by direct current or alternating current) during the annealing process [32,52,53,54]. The combined influence of the stress field and magnetic field on the thermal diffusion behavior of magnetic atoms leads to atomic reorientation, thereby inducing MA. The magnitude and direction of the applied magnetic field can be easily adjusted, with common configurations including longitudinal, transverse, and rotating magnetic fields [55,56]. Studies have shown that the MA induced in Fe-based and Co-based alloys through stress magnetic field annealing differs significantly from the effects of stress or magnetic field alone under the same annealing conditions. This additional MA is attributed to the ordering of atomic pairs induced by the magnetic field, superimposed on the stress-induced MA.

Compared to conventional SA, the MA induced by stress magnetic field annealing has a more complex physical mechanism. The magnetic field induces uniaxial MA, which is closely related to the ordered arrangement of atoms along the local magnetization direction. This arrangement minimizes the spin–orbit coupling energy of the system. Figure 8 illustrates the hysteresis loops and corresponding magnetic parameters of Fe-based nanocrystalline alloys treated with differently oriented magnetic fields [57,58]. Flat hysteresis loops (F1 and F2) are obtained through transverse magnetic field annealing, where magnetization involves the rotation of the magnetization vector from the easy axis to the hard axis, with domain rotation being the dominant process. In contrast, a rectangular hysteresis loop (Z) is obtained through longitudinal magnetic field annealing, which induces MA parallel to the ribbon axis, with domain wall motion being the dominant process. A circular hysteresis loop (R) is observed when annealing is performed without an applied magnetic field, indicating that the magnetization process is dominated by a mixture of domain rotation and domain wall motion. It is important to note that annealing without a magnetic field does not preclude the induction of MA. As long as the annealing temperature is below the Curie temperature of the nanocrystals, MA is always induced along the direction of local spontaneous magnetization. This type of MA is referred to as magnetization-induced MA. Rotating magnetic field annealing can significantly reduce coercivity and enhance remanence, thereby further reducing the fluctuations of these magnetization-induced MAs [57].

In a word, SA combined with a magnetic field offers a versatile approach for optimizing the magnetic properties of soft magnetic materials. This technique not only enhances the material’s performance but also provides a platform for tailoring MA and microstructural refinement, making it a promising strategy for developing high- performance soft magnetic materials [52,53,54,55,56].

## 4. Influence of Annealing Parameters on Magnetic Anisotropy

Annealing is a critical process that modifies the microstructure of a material to regulate its magnetic properties. This process involves several physical mechanisms, including structural relaxation, atomic rearrangement, phase transformation, and the release of residual internal stresses. By carefully tuning annealing parameters such as temperature, time, stress, and heating or cooling rates, the shape of the hysteresis loop can be effectively adjusted.

The annealing temperature plays a pivotal role in determining the magnitude and nature of MA. For Fe-based alloys, SA below the crystallization temperature typically induces a smaller MA compared to annealing above this temperature. However, if the annealing temperature is significantly higher than the crystallization temperature (reaching the second crystallization stage), the soft magnetic properties of the material can deteriorate severely. This deterioration is primarily due to the precipitation of intermetallic compounds (e.g., Fe_2_B or Fe_3_B phases in Fe_73.5_Cu_1_Nb_3_Si_9_B_13.5_ alloy at annealing temperatures above 600 °C, with K_1_ = 430 kJ/m^3^) and grain coarsening [57]. Therefore, the optimal annealing temperature must be carefully selected to fall within an appropriate range above the crystallization temperature. For example, the optimized annealing temperature for conventional stress-annealed FINEMET-type alloys is around 550 °C. As shown in Figure 9, for FINEMET-type alloys annealed below 480 °C (with the same annealing time and stress), nearly zero MA parallel to the stress direction is induced. When the annealing temperature reaches or exceeds 480 °C, the sign of MA changes, and its value increases with increasing annealing temperature, reaching a maximum in the range of 500–560 °C [38].

To ensure sufficient structural relaxation, appropriate annealing time is essential for SA. In the case of stress-annealed FINEMET-type alloys, the time required for MA to reach saturation decreases with increasing annealing temperature. At 540 °C, if the annealing time exceeds 10–20 min [59], the induced MA will reach a saturated value and remain essentially unchanged. In other words, the soft magnetic nanocrystalline structure is essentially fully formed after SA for 10–20 min. Within this range, the stress-induced MA becomes relatively insensitive to annealing time and temperature after the nanocrystalline structure is formed [59].

The annealing stress also has a significant effect on MA. As shown in Figure 9 [38], under the same annealing conditions, the induced MA increases with increasing annealing tensile stress. Moreover, the sensitivity of MA to stress increases with increasing annealing temperature and reaches a maximum at the optimized annealing temperature. This suggests that the interplay between stress and temperature is crucial for maximizing the induced MA [38].

In addition, the heating and cooling rates also affect the thermodynamics and kinetics of crystallization. A uniform and rapid cooling rate is conducive to obtaining a uniform nanobiphasic heterogeneous structure. This is particularly important for maintaining the desired soft magnetic properties and ensuring the formation of a stable nanocrystalline structure.

In summary, the MA of Fe-based soft magnetic materials can be effectively modulated by carefully optimizing annealing parameters such as temperature, time, stress, and heating (cooling) rates. A well-balanced combination of these parameters is essential for achieving the desired MA and magnetic properties, thereby meeting specific application requirements. Future research should focus on further elucidating the underlying mechanisms and developing more precise annealing protocols to enhance the performance of soft magnetic materials.

## 5. Influence of Alloy Composition on Magnetic Anisotropy

The design and optimization of amorphous soft magnetic alloys have been extensively studied to enhance their soft magnetic properties through tailored chemical compositions and SA-induced MA. The MA is significantly influenced by both annealing conditions and alloy compositions [58,60]. Typically, Fe-based alloys exhibit positive magnetostriction coefficients (λs = 20–40 ppm), while Co-based alloys display negative magnetostriction coefficients (λs = −5–−3 ppm). Many studies have shown that the magnetostriction coefficient is closely related to alloy composition. For example, adding small amounts of Co to FeZrB alloy systems or incorporating Fe and Mn into Co-based alloys can yield materials with near-zero magnetostriction coefficients [60]. A typical composition for such alloys is (Co_0.95_Fe_00.5_)_100−x_Mn_x_ [61]. Moreover, the magnetostriction coefficient can be regulated by optimizing the alloy structure. For instance, growing a biphasic heterogeneous structure from the amorphous precursor of Fe-based alloys, such as Fe_73.5_Cu_1_Nb_3_Si_13.5_B_9_, can reduce the saturation magnetostriction coefficient. This can be described by the balance of the two-phase structure [62]:(2)$$\lambda_{s}=p\lambda_{s}^{\prime }+(1-p)\lambda_{s}^{\prime \prime },$$
where *p* is the volume fraction of the α-Fe(Si) nanocrystalline phase, and *λ*′_s_ and *λ*″_s_ are the saturation magnetostriction coefficients of the α-Fe(Si) nanocrystalline phase and the residual amorphous matrix phase, respectively. To achieve a near-zero magnetostriction coefficient in Fe-based alloys, it is necessary to increase the crystallization volume fraction *p* of the α-Fe(Si) nanocrystalline phase with a negative magnetostriction coefficient. Recent studies on the intrinsic structure of amorphous alloys have revealed that they possess short- and medium-range ordered atomic clusters and long-range disorder structures [63,64,65]. Similar to nanobiphasic structures, the magnetostriction coefficient of amorphous alloys can also be represented by Equation (2).

Table 2 shows the saturation magnetostriction coefficients and the MA (denoted as K) for various compositions of nanocrystalline (NA) and amorphous (AM) alloys under different annealing conditions. For example, in FINEMET-type alloys, the MA largely depends on the Si concentration in the α-Fe(Si) nanocrystalline phase. Alloys with high Si concentrations (greater than 10 at.%) exhibit MA perpendicular to the stress direction (hard axis), while those with low Si concentrations (less than 10 at.%) [62] show MA parallel to the stress direction (easy axis). This indicates a change in the direction of MA at a Si content of 10%. When Co is substituted for part of the Fe in FINEMET alloys, such as in the Fe_73.5−x_Co_x_Cu_1_Nb_3_Si_15.5_B_7_ (x = 0, 10, 20, 30 at.%) alloy series, the addition of Co reduces the Si concentration in the α-Fe(Si) phase. Transverse MA is observed when x = 0 and x = 10, while longitudinal MA is induced when x = 20 and x = 30 [66]. This suggests that the change in the easy magnetization axis of MA is related to the sign of the saturation magnetostriction coefficient of the Fe-Si-Co solid solution phase (from negative to positive). Müller et al. [67] replaced a significant portion of Fe in Fe_73.5_Cu_1_Nb_3_Si_13.5_B_9_ and Fe_86_Zr_7_B_6_Cu_1_ alloys with Co, achieving a high induced MA while increasing the saturation magnetostriction coefficient by an order of magnitude. Adding Cr to FINEMET alloys can enhance the thermal stability of MA and improve the corrosion resistance of the alloy. For instance, in the Fe_73.5−x_Cr_x_Cu_1_Nb_3_Si_13.5_B_9_ (x = 1, 2, 3, 5) alloys, the induced transverse MA decreases with increasing Cr content [68,69]. The largest reported induced MA (approximately 50 kJ/m^3^) is found in the Co_72_Fe_4_Mn_4_Nb_4_Si_2_B_14_ alloy. For amorphous alloys (such as Fe-, Co-, Ni-, and FeNi-based alloys), the induced MA is one to two orders of magnitude lower than that of nanocrystalline alloys.

In a word, the influence of alloy composition on MA is primarily reflected in the changes in the saturation magnetostriction coefficient. In addition, the size and morphology of nanocrystalline grains also have an impact on MA. The former can be understood from the perspective of size effects, while the latter can be interpreted in terms of shape anisotropy. Generally, under the same annealing conditions, the induced MA in Co-based nanocrystalline alloys is significantly higher than that in Fe-based and other alloys, and its magnitude is positively correlated with the saturation magnetostriction constant λs. Alloys containing more than two magnetic elements, such as Fe-Co-Ni-based alloys, usually induce stronger MA than those containing only one magnetic element, such as pure Fe-based alloys. However, these statements are qualitative, and whether the MA induced by SA is necessarily related to the saturation magnetostriction coefficient remains a scientific issue worthy of further discussion.

## 6. Physical Mechanisms of Stress-Induced Magnetic Anisotropy

Despite extensive research on MA as a critical factor influencing magnetic properties, its rational regulation can significantly enhance the soft magnetic performance of materials. Understanding the formation mechanisms of SA-induced MA is crucial for designing high-performance soft magnetic alloy materials. However, the formation mechanism of SA-induced MA in Fe-based amorphous and nanocrystalline alloys remains controversial. To date, three well-known models have been proposed: (1) the atomic pair ordering model proposed by Hofmann et al. [74], (2) the MEC model based on the magnetostriction effect proposed by Herzer [62], and (3) the distribution anisotropy model based on the preferential clustering of nanocrystalline grains proposed by Fang et al. [75]. The following sections will elaborate on the physical ideas and formation processes of these three models.

### 6.1. Atomic Pair Ordering Model

In the 1950s, Néel [76] proposed the concept of atomic pair ordering in binary alloy systems, assuming that the energy of each atomic pair (A-B, A-A, B-B) depends not only on the angle (θ) between the orientation of the atomic pair and the local magnetization (M), but also on the interatomic distance (r). When an alloy is heated to a sufficiently high temperature, the atoms rearrange. If a magnetic field is applied during this process, the system forces atomic pairs with different orientations to preferentially align along the magnetic field direction to minimize the spin–orbit coupling energy [77,78]. Upon cooling to room temperature, this preferentially oriented atomic pair arrangement is “frozen”, ultimately forming field-induced MA. In the 1990s, Hofmann et al. [58,74] used the concept of atomic pair ordering to explain the SA-induced MA in multiphase materials such as Fe_73.5_Cu_1_Nb_3_Si_9_B_13.5_. Assuming that the MA energy is confined only to the α-Fe(Si) nanocrystalline grains and neglecting the contribution of the residual amorphous matrix, the SA-induced MA can be expressed as follows:(3)$$K=\frac{33}{64}v\frac{c_{A}^{2}c_{B}^{2}L_{0}\sigma D_{0}}{C_{3}\kappa RT_{a}r_{0}},$$
where *v* is the volume fraction of the α-Fe(Si) nanocrystalline phase; *c*_A_ and *c*_B_ are the atomic concentrations of Fe and Si, respectively; *L*_0_ = *N*_A_*l*_0_, where *l*_0_ is the local MA of each atomic pair; *D*_0_/*r*_0_ is the change in the average interatomic distance related to the macroscopic deformation of the material caused by atomic rearrangement during SA; *C*_3_ = *C*_44_ is the elastic constant; *R* is the gas constant; *k* is the compressibility coefficient; *T*a is the annealing temperature; and *σ* is the annealing tensile stress. Since the theoretical calculation values are of the same order of magnitude as the experimental observations, Hofmann et al. [58,74] suggested that SA-induced MA may originate from the ordering of Fe-Si atomic pairs in α-Fe(Si) nanocrystals, whose physical model is described in Figure 10. However, this model faces challenges in explaining the relationship between MA and Si concentration. Ohnuma et al. [14] used transmission X-ray diffraction to discover that LPA in nanocrystals is the structural origin of stress-induced MA and pointed out that LPA is unrelated to the Si concentration in α-Fe(Si) nanocrystals. Therefore, the experimental results of Ohnuma et al. [14] negated the contribution of Fe-Si atomic pair ordering to MA.

### 6.2. Magneto-Elastic Coupling Model

In Fe-based amorphous alloys, Kraus et al. [32] pointed out that the viscoelastic polarization between amorphous phases is the primary source of SA-induced MA [79,80,81,82]. Based on the viscoelastic theory, Herzer [62] proposed that SA-induced MA originates from magneto-elastic anisotropy and defined the virtual magnetostriction coefficient k as follows:(4)$$k=-\frac{2K}{3\sigma },$$

Here, *σ* is the stress applied during heat treatment. Additionally, the normalized virtual magnetostriction coefficient (*k*/*v*_cr_) shows a highly consistent trend with the saturation magnetostriction coefficient as the Si content varies in Fe-based alloys. Combining the Equation (4), stress-induced MA can be expressed as follows:(5)$$K=-\frac{3}{2}v_{cr}\lambda_{s}^{FeSi}\sigma ,$$
where *v*_cr_ is the volume fraction of the α-Fe(Si) nanocrystalline phase. Herzer [62] suggested that stress-induced MA mainly stems from the MEC between the internal stress imposed on the α-Fe(Si) crystals by the anelastic deformation of the amorphous matrix and its saturation magnetostriction coefficient ($\lambda_{s}^{FeSi}$). This mechanism is illustrated in Figure 11 [62,83]. Varga et al. [67] supported the MEC model as the primary physical mechanism for SA-induced MA based on the correlations among MA, dislocation mobility, and magnetostriction coefficients in different Fe-based nanocrystalline alloys. To obtain experimental evidence for the structural origin, Ohnuma et al. [14] used transmission X-ray diffraction to observe the microstructure of stress-annealed Fe-based nanocrystalline alloys. They found that the peak positions (2θ angles) of the (310) diffraction peaks differ by 0.1° between the parallel and perpendicular stress directions, corresponding to a 0.2% lattice spacing difference. However, this difference was not observed in the absence of tensile stress. The experimental results indicated that the lattice spacing of α-Fe(Si) nanocrystals increases along the stress direction and decreases along the perpendicular direction, leading to LPA (schematic diagrams shown in Figure 12), providing direct evidence for the structural origin of stress-induced MA. Subsequent studies revealed a linear relationship between the lattice strain of α-Fe(Si) nanocrystals and stress-induced MA, indicating that this strain is elastic. Consequently, Herzer’s MEC model was refined as follows [66]:(6)$$K=-\frac{3}{2}v\lambda_{s}^{FeSi}E\varepsilon ,$$
where *E* and ε are the Young’s modulus and microstrain of the α-Fe(Si) nanocrystalline phase, respectively. The experimental results confirmed that the magneto-elastic coupling model is the primary mechanism for SA-induced MA in Fe-based nanocrystalline alloys.

Despite its widespread acceptance, the limitations of the MEC model have also been revealed. Fang et al. [15,16,17,84] proposed a method for in situ observation of microstructures using synchrotron radiation X-ray diffraction, elevating the study of SA-induced MA to an in situ, dynamic perspective. The experimental results showed that the macroscopic strain is negative during free annealing, primarily due to the contraction of the alloy ribbon caused by atomic rearrangement during nanocrystallization. In contrast, the macroscopic strain is positive during SA, mainly due to the collective macroscopic behavior of the elastic strain of nanocrystals and the creep of the amorphous matrix. Additionally, free-annealed nanocrystals exhibit isotropic growth without significant LPA, whereas stress-annealed nanocrystals show pronounced LPA, with different kinetics in the parallel and perpendicular directions during nanocrystallization [15,16,17,84]. Furthermore, they systematically investigated the magnetic anisotropy energy (MAE) and 3d orbital density of states (DOS) characteristics of Fe atoms through comprehensive first-principles computational analysis. Their results demonstrated that applied strain dynamically modulates the spin–orbit coupling (SOC) interactions in the 3d orbitals of FeI and FeII atoms near the Fermi level, particularly those contributing to the electronic states at the energy frontier. This orbital-specific SOC modification mechanism was shown to significantly enhance the MAE values. The study provides compelling evidence that the emergence of stress-induced MA fundamentally originates from strain-mediated modifications in the 3d orbital SOC parameters of Fe atoms [84], and the related results are shown in Figure 13.

However, Fang et al. [37,85] further predicted the relaxation kinetics of LPA from both experimental and theoretical perspectives, revealing that MA is not fully reversible, and the results are shown in Figure 14. This finding challenges the MEC model, which suggests that tempering can completely eliminate LPA contributed by lattice elastic strain. Therefore, the MEC model may not be the only mechanism for stress-induced MA.

### 6.3. Distribution Anisotropy Model

At the mesoscale, Fang et al. [75] observed the directional clustering of nanocrystals in stress-annealed Fe-based nanocrystalline ribbons using atomic force microscopy and proposed that the distribution anisotropy of α-Fe(Si) nanocrystals is the physical mechanism underlying stress-induced MA. During annealing, tensile stress causes the alloy ribbon to stretch axially while contracting laterally. From a microscopic perspective, this can be understood as a change in the average interatomic distance in α-Fe(Si) nanocrystals, with an increase in the direction parallel to the stress and a decrease in the direction perpendicular to the stress [75]. The crystallization process of Fe-based alloy ribbons under tensile stress can be simply understood as the repositioning of non-equilibrium atoms near their equilibrium positions. Additionally, due to the damping effect of large atoms like Nb, some atoms fail to return to their equilibrium positions, resulting in the coexistence of amorphous and nanocrystalline phases [75]. Since the amorphous phase has a lower elastic modulus, it is more prone to deformation under stress, causing the amorphous phase between nanocrystals to stretch in the direction parallel to the stress and contract in the direction perpendicular to the stress. Assuming that the deformation of the nanocrystalline phase (such as α-Fe(Si) nanocrystals) is much smaller than that of the amorphous phase (such as the residual amorphous matrix in Fe-based nanocrystalline alloys), the distribution of nanocrystals becomes anisotropic, primarily originating from the inelastic deformation of the amorphous matrix [75]. The experimental results and the schematic diagrams of the distribution anisotropy model are shown in Figure 15. However, the experimental evidence for this model is not yet fully sufficient, as the preparation process of the alloy ribbon cross-section may lead to atomic surface reconstruction [75]. Based on this observation, Fang et al. proposed that stress-induced MA originates from the synergistic effect of LPA and the distribution anisotropy of nanocrystals, and the MAE can be described as follows [37,75]:
(7)$$K=K_{e}+K_{d},$$
where *K*_e_ is the contribution of LPA to MA, and *K*_d_ is the contribution of nanocrystal distribution anisotropy to MA. *K_d_* = *k*Δ*δ*, where Δ*δ* represents the anisotropy of nanocrystalline grain distribution and *k* is the coupling coefficient.

Although the MEC model has made significant progress in explaining SA-induced MA, its limitations have prompted researchers to explore new mechanisms. The distribution anisotropy model of nanocrystals offers a new direction for research in this field. Future studies need to further validate and integrate these mechanisms to gain a more comprehensive understanding of the nature of SA-induced MA.

## 7. Summary and Perspectives

This review systematically examines stress-induced MA in Fe-based amorphous and nanocrystalline soft magnetic alloys. The key findings and unresolved challenges are summarized as follows:(1)MA induction via SA (conventional, rapid, and hybrid magnetic-field-assisted methods) is highly sensitive to annealing parameters (temperature, time, stress magnitude, and heating/cooling rates). The precise control of these parameters enables tailored MA for optimized soft magnetic performance.(2)Alloy composition critically governs MA. Strategic doping (e.g., Co, Mn, Cr) modulates the magnetostriction coefficient and enhances MA. Optimal soft magnetism arises from balancing the α-Fe(Si) nanocrystalline phase and residual amorphous matrix, ensuring near-zero magnetostriction and minimized energy losses.(3)Three main models have been proposed to explain stress-induced MA: the atomic pair ordering model (limited in explaining the relationship between MA and Si concentration), the MEC model (which may not fully account for the irreversibility of MA), and the distribution anisotropy model (which requires further experimental validation). The distribution anisotropy model provides a new perspective by considering the preferential clustering of nanocrystals and the inelastic deformation of the amorphous matrix.

To advance the understanding of stress-induced MA and guide the design of high-performance soft magnetic materials, future research should prioritize the following directions:(1)Clarifying unresolved debates, including the reversibility of stress-induced MA, the equivalence between internal stress and macroscopic annealing stress, and the correlation between MA and saturation magnetostriction coefficient, is essential to unravel the stress-sensitive mechanisms in Fe-based alloys.(2)Integrating theoretical modeling, computational simulations (e.g., molecular dynamics or phase-field methods), and experimental validation will bridge atomic-scale mechanisms to macroscopic magnetic properties.

In summary, addressing these challenges through innovative methodologies and cross-scale analyses will accelerate the development of next-generation soft magnetic alloys with tailored anisotropy and enhanced performance.

## Figures and Tables

**Figure 1 materials-18-01499-f001:**
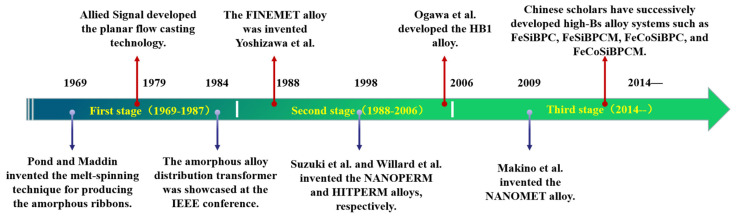
History of research and development of Fe-based soft magnetic alloys.

**Figure 2 materials-18-01499-f002:**
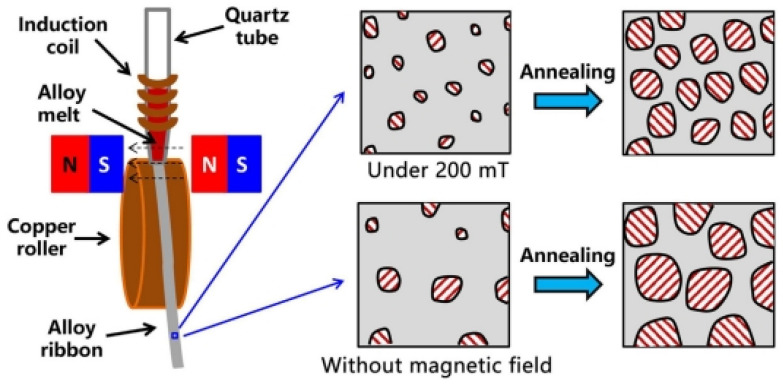
A schematic diagram of the production process for Fe-based nanocrystalline alloys. (Jia, X.J. 2023 [21]).

**Figure 4 materials-18-01499-f004:**
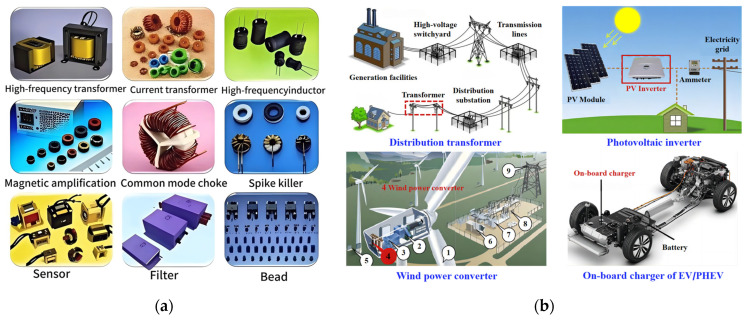
Fe-based amorphous and nanocrystalline soft magnetic alloy cores and their applications: (**a**) various cores; (**b**) the applications of magnetic cores in distribution transformers, photovoltaic inverters, wind power inverters, and onboard batteries (Li, F.C. 2019 [25]).

**Figure 5 materials-18-01499-f005:**
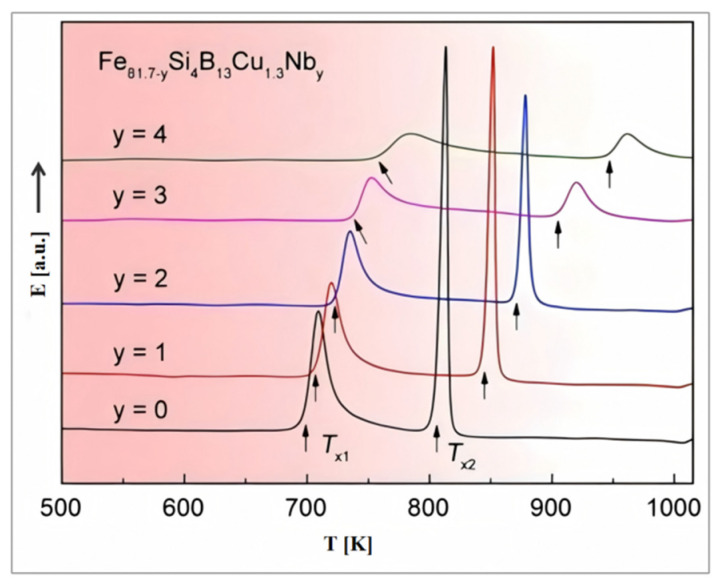
DSC curves of Fe_81.7−y_Si_4_B_13_Cu_1.3_Nb_y_ (y = 0, 1, 2, 3, 4) alloys. (Li, Y.H. 2021 [30]).

**Figure 6 materials-18-01499-f006:**
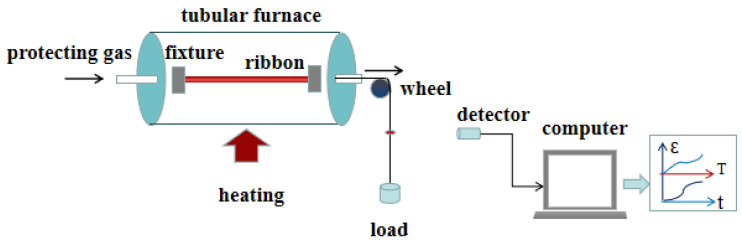
Schematic diagram of a conventional SA device.

**Figure 7 materials-18-01499-f007:**
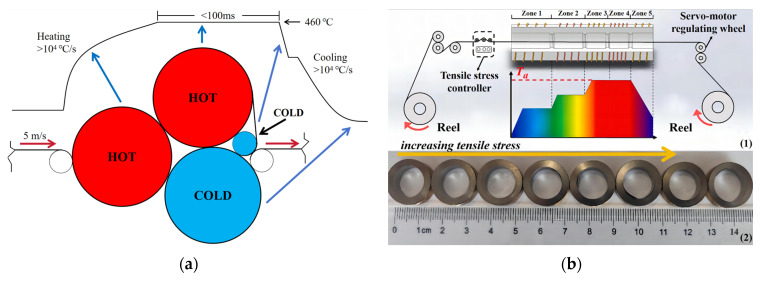
Schematic diagram of two types of rapid stress annealing devices for: (**a**) amorphous; (**b**) nanocrystalline, in Figure 7b, (1) illustrates a schematic diagram of a continuous stress annealing device, and (2) shows Fe-based nanocrystalline magnetic cores subjected to different stress annealing treatments (Zhu, F. 2023 [45], Francoeur, B. 2012 [51]).

**Figure 8 materials-18-01499-f008:**
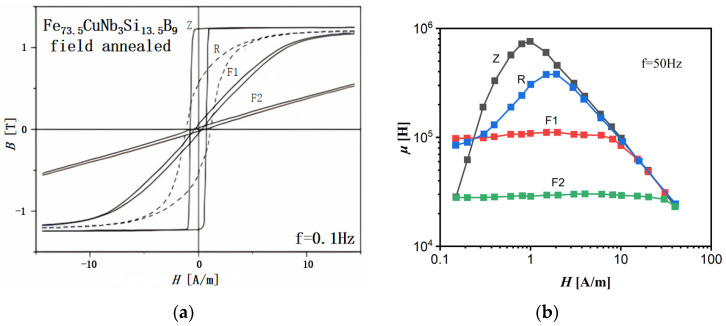
Magnetic properties of Fe-based nanocrystalline alloys treated with differently oriented magnetic fields: (**a**) hysteresis loops; (**b**) magnetic permeability, where Z, F1 (and F2), and R represent the Fe-based nanocrystalline alloys annealed with a longitudinal magnetic field, a transverse magnetic field, and no magnetic field, respectively. (Herzer, G. 1996 and 2013 [57,58]).

**Figure 9 materials-18-01499-f009:**
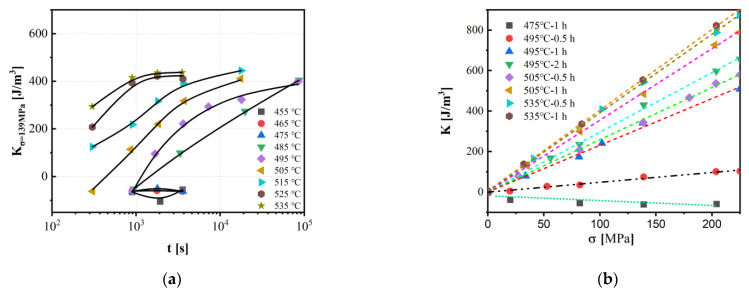
The curve of MA versus annealing process parameters: (**a**) annealing time; (**b**) annealing stress. (Alves, F. 2000 [38]).

**Figure 10 materials-18-01499-f010:**
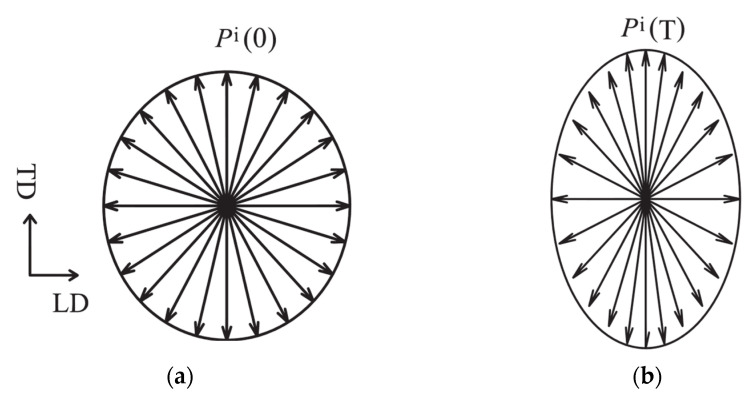
Schematic diagram of the atom-pair ordering model: (**a**) isotropic distribution; (**b**) anisotropic distribution. LD denotes the parallel stress direction, TD denotes the perpendicular stress direction, and *P*^i^(0) and *P*^i^(T) denote the distribution probability of atomic pairs in a certain direction without stress annealing and with stress annealing, respectively.

**Figure 11 materials-18-01499-f011:**
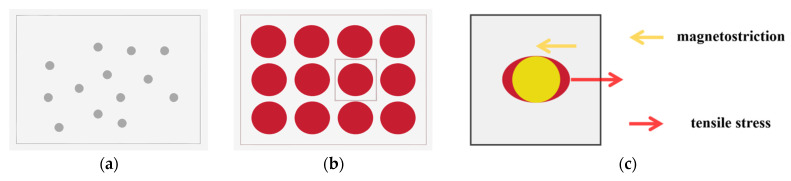
Schematic diagrams of the magneto-elastic coupling model: (**a**) and (**b**) represent the tensile-stressed annealed structures of amorphous and nanocrystalline alloy ribbons, respectively; (**c**) a magnified view of the boxed area in Figure 11b. The red circles represent the nanocrystals, the red arrows represent the tensile stresses due to the creep of the amorphous substrate, and the yellow arrows represent the shrinkage of the nanocrystals due to magnetostriction. (Herzer, K. 1994 [62], Csizmadia, E. 2015 [83]).

**Figure 12 materials-18-01499-f012:**
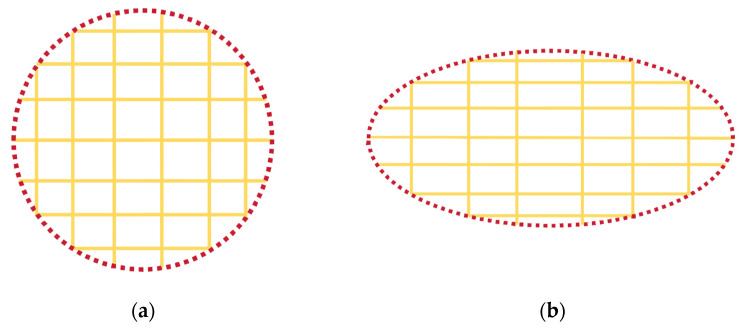
The schematic diagrams of anisotropy lattice plane distance: (**a**) free annealing; (**b**) stress annealing. The dark red dashed circles and ellipses represent the nanocrystals in the free-annealed and stress-annealed states, respectively, while the yellow lines indicate the crystallographic planes (hkl) of the nanocrystals. (Ohnuma, M. 2003 [14]).

**Figure 13 materials-18-01499-f013:**
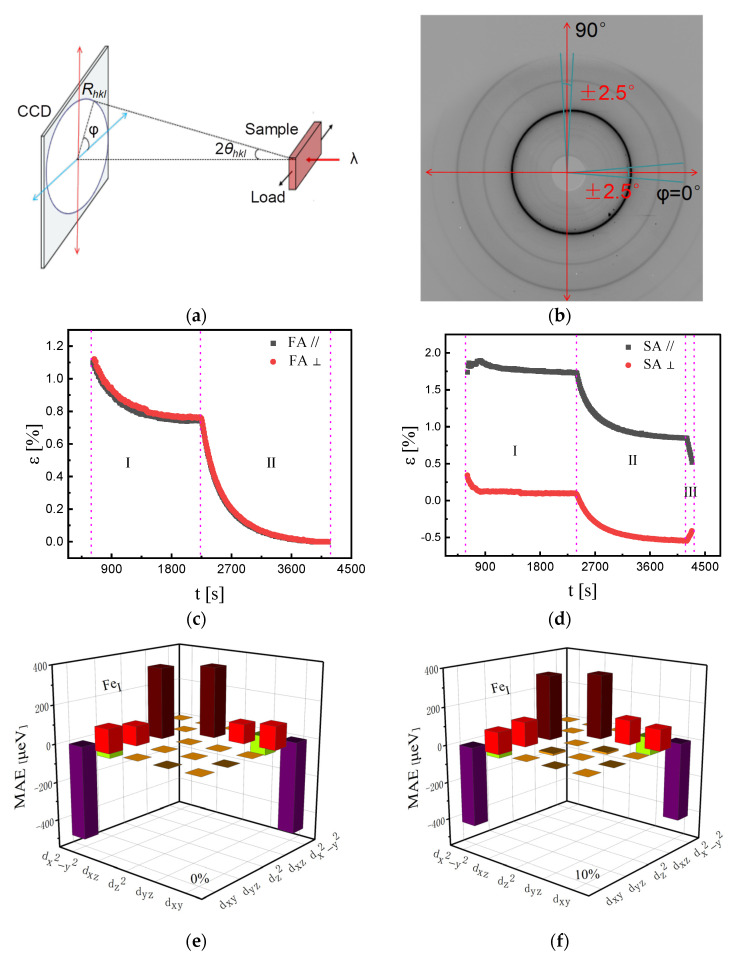
Results from synchrotron radiation in situ X-ray diffraction tests and first-principles calculations: (**a**) schematic diagram of the experimental principle; (**b**) 2D X-ray diffraction pattern obtained from CCD in (**a**); (**c**) and (**d**) microstrains of isotropic and anisotropic evolution in α-Fe(Si) nanocrystals annealed by free annealing (FA) and stress annealing (SA), respectively; (**e**) and (**f**) Fe_I_ atom 3d orbital resolution MAE in α-Fe(Si) under strains of 0% and 10%, respectively; (**g**) and (**h**) Fe_II_ atom 3d orbital resolution MAE in α-Fe(Si) under strains of 0% and 10%, respectively. Here, in (**e**–**g**), the different colors represent the contributions of orbital-pair MAE among the 5 sub-orbitals of iron atoms. (Fang, Y.Z. 2024 [84]).

**Figure 14 materials-18-01499-f014:**
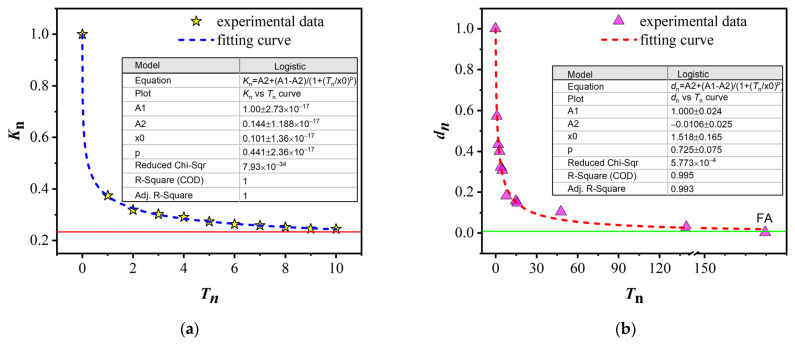
Relaxation dynamic curves: (**a**) MA; (**b**) LPA. Here, *T*n denotes the tempering frequency, while *K*n and *d*n represent the normalized magnetic anisotropy and normalized lattice plane anisotropy. Specifically, in (**a**), the red line marks the value at which the MA relaxation reaches a steady state, while the green lines in (**b**) denote the value at which the LPA relaxation reaches a steady state (Fang, Y.Z. 2022 [37]).

**Figure 15 materials-18-01499-f015:**
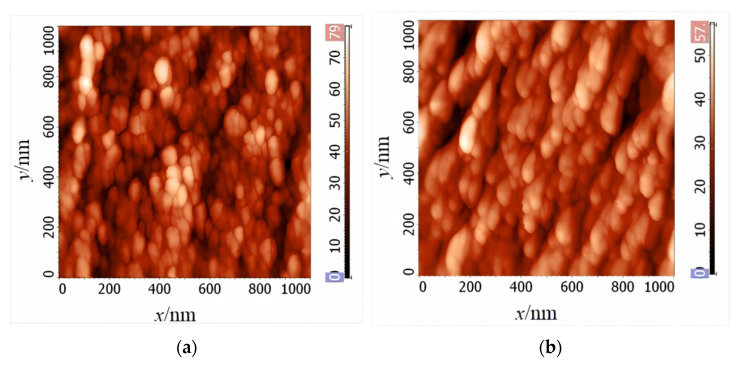
The experimental results and the schematic diagrams of distribution anisotropy model: (**a**) and (**b**) represent the cross-sectional topography of the Fe-based alloy ribbons in the free-annealed and stress-annealed states, respectively; (**c**) and (**d**) depict the distribution diagrams of the nanograins, where the gray solid spheres represent the nanocrystalline grains within the alloy strip, with the *x*-axis and *y*-axis denoting the directions parallel and perpendicular to the applied stress, respectively. (Fang, Y.Z. 2010 [75]).

**Table 1 materials-18-01499-t001:** Several typical Fe-based nanocrystalline alloys and their performance indices [18].

Alloys	Composition [at %]	Bs [T]	Hc [Am^−1^]	μ (1 kHz)	Tc [K]	D [nm]
FINEMET	Fe_73.5_Si_13.5_B_9_Nb_3_Cu_1_	1.24	0.53	100,000	573	10~12
NANOPERM	Fe_91_Zr_7_B_2_	1.70	7.2	14,000	~750	10~20
HITPERM	Fe_44_Co_44_Zr_7_B_4_Cu_1_	1.61	10	1800	980	10~17
NANOMET	Fe_85_Si_2_B_8_P_4_Cu_1_	1.85	5.8	27,000	728	~20

Note: Tc denotes the crystalline temperature, D denotes the grain size, and μ represents the magnetic permeability measured under a magnetic field at 1 kHz.

**Table 2 materials-18-01499-t002:** The MA induced in alloys of different compositions under certain annealing conditions.

States	Types	Alloy Compositions	Annealing Conditions			Magnetostriction	MA
			*T_a_* [°C]	*t* [min]	σ [MPa]	*λ*_s_ [ppm]	*K* [kJ/m^3^]
NA	FINEMET [58,70,71,72]	Fe_77.5_Cu_1_Nb_3_B_15.5_	450	20	433	−0.60	-
		Fe_73.5_Cu_1_Nb_3_Si_9_B_13.5_	550	20	469	−3.30	2.000
		Fe_73.5_Cu_1_Nb_3_Si_15.5_B_7_	550	20	600	−7.70	7.000
			700	5	54		0.5963
			700	5	200		2.313
			700	5	342		4.356
			675	5	19.2		0.1913
			675	5	204		2.294
			675	5	339		4.188
			675	15	0		0.0063
			675	15	105.4		1.138
			675	15	132.5		1.45
		Fe_73.5_Cu_1_Nb_3_Si_13.5_B_9_	550	60	0		0.0227
			550	60	20		0.2168
			550	60	41		0.4534
			550	60	62		0.622
			550	60	82		0.8929
			550	60	102		1.152
			540	60	850	-	8.000
	Co-FINEMET [46]	Fe_68.5_Co_5_Cu_1_Nb_3_Si_15.5_B_7_	580	4	250	−10.0	2.900
	Cr-FINEMET [68]	Fe_72.5_Cr_1_Cu_1_Nb_3_Si_13.5_B_9_	520	120	150	-	2.500
		Fe_71.5_Cr_2_Cu_1_Nb_3_Si_13.5_B_9_	520	120	150	-	2.380
		Fe_70.5_Cr_3_Cu_1_Nb_3_Si_13.5_B_9_	520	120	150	-	2.250
		Fe_69.5_Cr_4_Cu_1_Nb_3_Si_13.5_B_9_	520	120	150	-	2.120
	Co-based [41,73]	Co_89_Zr_7_B_4_	550	4	300/250	−27.0/−26.0	9.000/8.000
		(Co_97.5_Fe_2.5_)_89_Zr_7_B_4_	550	4	300/250	−30.0/−28.0	11.000/8.000
		(Co_90_Fe_10_)_89_Zr_7_B_4_	550	4	300/250	−32.0/−30.0	12.000/9.000
		(Co_97.5_Fe_2.5_)_89_Zr_7_B_4_	550	4	450	−34.2	18.900
		Co_72_Fe_4_Mn_4_Nb_4_Si_2_B_14_	520	30	200/600	1.62/-	16.700/50.000
AM [54]	Co-based	Co_69.5_Fe_3.5_Mo_3_Nb_1_Si_16_B_7_	350	0.5	249	−0.06	0.209
		Co_72_Fe_1_Mn_4_Mo_1_Si_13_B_9_	350	0.5	254	−0.12	0.324
		Co_72.5_Fe_5_Si_5.5_B_17_	350	0.5	353	−0.25	0.616
		Co_72.5_Fe_1.5_Mn_4_Si_5_B_17_	350	0.5	332	−0.41	0.424
		Co_56.5_Fe_6_Mn_1_Ni_16_Si_4_B_16.5_	350	0.5	364	−0.90	1.406
	FeNi-based	Fe_24_Ni_58_Mo_2_B_16_	350	0.5	308	7.50	2.666
	Ni-based	Fe_24_Co_12_Ni_46_Si_2_B_16_	350	0.5	293	10.9	1.626
	FeNi-based	Fe_40_Ni_40_Mo_4_B_16_	350	0.5	328	14.5	2.865
	Fe-based	Fe_65_Co_18_Si_1_B_16_	350	0.5	336	42.9	−1.053

Note: Ta denotes the annealing temperature, t denotes annealing time, σ denotes annealing stress, and K denotes the density of magnetic anisotropy energy.

## Data Availability

The data presented in this study are available on request from the corresponding author.

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
