# Peer review of "Stress-Induced Magnetic Anisotropy in Fe-Based Amorphous/Nanocrystalline Alloys: Mechanisms, Advances and Challenges"

_materials, 2025, doi:10.3390/ma18071499_

Round 1
Reviewer 1 Report
Comments and Suggestions for Authors
This work presents the stress-induced magnetic anisotropy in Fe-based amorphous/ nanocrystalline alloys: mechanisms, advances and challenges. The paper is acceptable for publication in the present form but some suggestions should be considered by the authors:
- The authors discuss the hysteresis loops and corresponding magnetic parameters of Fe-based nano-crystalline alloys treated with different oriented magnetic fields. The authors claim that “Flat hysteresis loops (F1 and F2) are obtained through transverse magnetic field annealing, where magnetization involves rotation of the magnetization vector from the easy axis to the hard axis, with domain rotation being the dominant process.” What is the physical meaning of “domain rotation”? What happen into domains?
- The authors should make cross-section of the Fe-based nano-crystalline alloys to investigate the microstructure of the samples.
Author Response
Dear Reviewer,
Thank you very much for your constructive comments on our manuscript. We appreciate your insights and have carefully considered your suggestions. Below are our detailed responses and the actions we have taken to address your concerns:
Regarding the Physical Meaning of "Domain Rotation" You have raised a question about the term “domain rotation” in our discussion of the hysteresis loops and corresponding magnetic parameters of Fe-based nanocrystalline alloys treated with different oriented magnetic fields. Specifically, you asked about the physical meaning of "domain rotation" and what happens within the domains.
Response: "Domain rotation" refers to the process where the magnetization vector within magnetic domains rotates from the easy axis to the hard axis under the influence of an external magnetic field. This is a fundamental mechanism in magnetic materials, particularly those with significant magnetic anisotropy. Here is a more detailed explanation:
Magnetization Vector Rotation: In materials with magnetic anisotropy, the magnetization vector prefers to align along the easy axis due to lower energy states. When an external magnetic field is applied transverse to the easy axis, the magnetization vector must rotate to align with the field. This rotation involves overcoming energy barriers caused by magneto-crystalline anisotropy and magneto-elastic anisotropy.
Domain Wall Movement: During this process, domain walls (boundaries between regions of different magnetization directions) move to accommodate the change in magnetization direction. This movement is crucial for the overall magnetization process, especially in polycrystalline or nanocrystalline materials where domain structures are prevalent.
Energy Considerations: The rotation of the magnetization vector and the movement of domain walls involve energy changes. The energy required for this process depends on the strength of the magnetic anisotropy and the applied magnetic field. In our review, the flat hysteresis loops observed after transverse magnetic field annealing indicate that domain rotation is the dominant process, suggesting that the energy barriers for magnetization rotation are relatively low compared to other magnetization mechanisms (such as coherent rotation or nucleation).
Regarding Cross-Sectional Analysis of Microstructure You suggested that we perform cross-sectional analysis of the Fe-based nanocrystalline alloy samples to investigate their microstructures. We fully agree with this suggestion and have already conducted the corresponding cross-sectional analysis in our experiments.
Response: We have performed cross-sectional analysis of the Fe-based nanocrystalline alloy samples using Atomic Force Microscopy (AFM) and Magnetic Force Microscopy (MFM) to investigate their microstructures. These techniques provide valuable information on the cross-sectional topography, grain agglomeration, and fine magnetic domain structures, which are crucial for understanding the observed magnetic properties. Figure 1 show Results of cross-sectional characterization of the alloy ribbon using Atomic Force Microscopy (AFM) and Magnetic Force Microscopy (MFM).These results will be published in subsequent works.
Thank you again for your thoughtful review and suggestions. We believe that incorporating these improvements will significantly enhance the quality and clarity of our manuscript. We look forward to your further comments and feedback.
Best regards,
Jianqiang Zhang and Yanjun Qin

Reviewer 2 Report
Comments and Suggestions for Authors
The submitted manuscript is a review paper. The authors address the topic of Fe-based amorphous and nanocrystalline alloys, which are considered preferred dual-green energy-saving materials due to their unique magnetic properties, such as high magnetic permeability, low coercivity, and near-zero saturation magnetostriction.
As demonstrated in the reviewed paper, these properties have led to their widespread application in power electronics and information technology, including as magnetic core materials in high-frequency transformers, common mode chokes, ground fault interrupters, and motor rotors.
However, the authors of the manuscript also indicate that current devices utilizing Fe-based amorphous and nanocrystalline soft magnetic alloys do not fully reflect the advantages of these materials, primarily due to a lack of comprehensive understanding of their stress sensitivity.
The manuscript prepared by the authors provides a comprehensive review of research progress on stress-induced magnetic anisotropy in Fe-based amorphous and nanocrystalline alloys, focusing on five key areas: the development history of Fe-based alloys, heat treatment methods, annealing processes, chemical composition, and related physical mechanisms.
A review of the manuscript suggests that this manuscript constitutes a highly valuable reference for understanding the stress sensitivity of Fe-based alloys and will contribute to the development of green and efficient soft magnetic alloys.
The paper is interesting; however, like any article at the stage of preparation for publication, it contains some elements that need improvement.
In scientific papers, terms such as "study, manuscript, paper, scientific article, article," etc., should be used instead of "work." Please make this change in the "Funding" section of the manuscript.
The abstract is too long – please shorten it and avoid including results or descriptions of research methods. The abstract should encourage the reader to read the paper and indicate its topic – it should be a maximum of five to six sentences.
The introduction is appropriate and relatively brief. However, it lacks several graphics illustrating the current state of knowledge. Such graphics can be developed based on source papers or cited from them. Adding these will enhance the quality of the paper.
The manuscript should include nomenclature, which can be placed at the end of the paper. This should be a complete list of abbreviations, symbols, and notations used in the text.
All physical units in figures and tables should be written in square brackets, with a space after the symbol to which the unit refers. Units should not be written in parentheses or following a slash ("/") after the name of the quantity. It is recommended to introduce variable symbols in figures rather than writing out their full names. Please update the manuscript accordingly.
All graphs and figures that are not photographic images should be prepared in vector graphic quality. Currently, these figures are in raster quality and lose clarity when the document is zoomed in. Such presentation of results is not acceptable. Please make these changes if possible—perhaps the authors have source graphics that cannot be edited.
There are no comments regarding the references. The authors cite 85 scientific articles, effectively presenting the information contained within them.
Please review the entire manuscript for consistency in writing physical units. For example, "Mpa" appears in the text, but it should be "MPa." This is a simple typographical error but is noticeable and must be corrected.
In some cases, when presenting graphs, it would be beneficial to draw trend lines and provide regression equations along with determination coefficients. Please consider this option and try to incorporate this suggestion in the revised version of the manuscript.
The authors have structured the manuscript quite well and selected appropriate articles for review. The paper represents a significant contribution to the authors' field of study. It is well-organized, and the discussed issues are described correctly and without ambiguity.
The reviewed manuscript is scientifically sound and does not mislead the reader. The authors have appropriately selected and sufficiently referenced the literature.
The English used in the manuscript is understandable and does not raise any concerns regarding clarity.
The stuy has potential but requires some improvements – I suggest a minor revision.
Please implement the suggested revisions and resubmit the manuscript for further review.
Author Response
Dear Reviewer,
Thank you very much for your detailed review and valuable suggestions. We highly appreciate your feedback and have made the corresponding modifications and improvements based on your comments. Here are our responses to the specific issues you raised:
1. Regarding Terminology Usage
You pointed out that terms such as "study, manuscript, paper, scientific article, article" should be used instead of "work" in the "Funding" section. We have revised the relevant wording to ensure the use of terminology is more standardized.
- Regarding the Length of the Abstract
You mentioned that the abstract is too long and suggested shortening it while avoiding the inclusion of research results or method descriptions. We have condensed the abstract to ensure it summarizes the topic of the paper within five to six sentences, making it more engaging for readers to continue reading.You mentioned that the abstract is too long and suggested shortening it while avoiding the inclusion of research results or method descriptions. In response, we have condensed the abstract to ensure it summarizes the topic of the paper within five to six sentences, making it more engaging for readers to continue reading. The revised abstract is highlighted in red in the modified manuscript, as shown in the attachment.
3. Regarding the Addition of Figures in the Introduction
You suggested adding some figures to the introduction to better illustrate the current state of knowledge. We fully agree that this would significantly enhance the quality and readability of the manuscript. While we were unable to include the relevant figures in this submission due to time constraints, we have taken steps to streamline and refine the introduction. These revisions, which aim to improve clarity and conciseness, are highlighted in red within the revised introduction section of the manuscript. We will prioritize the inclusion of appropriate figures in future revisions, developing or citing them based on source papers. Thank you again for this valuable suggestion, which we will seriously consider in our subsequent work.
4.Regarding the Nomenclature
You suggested adding a nomenclature section at the end of the paper to list abbreviations, symbols, and notations used throughout the text. We have incorporated this suggestion by including a dedicated "Nomenclature" section (as shown in "Abbreviations") to help readers better understand the specialized terms and symbols used in the manuscript. This addition will enhance clarity and accessibility for readers who may be unfamiliar with some of the terminology.
5. Regarding the Format of Physical Units
You pointed out that physical units in figures and tables should be written in square brackets, with a space after the symbol to which the unit refers, rather than in parentheses or following a slash ("/"). We have updated all relevant figures and tables to ensure the units are presented in accordance with this standard. The changes are highlighted in red within the figures and tables, as shown in the revised manuscript.
- Regarding the Vector Graphic Quality of Figures
You mentioned that the current figures are raster images, which lose clarity when zoomed in, and suggested using vector graphics to maintain clarity. We have re-prepared the majority of the relevant figures to ensure they are presented in vector graphic format, meeting the publication requirements. However, a few figures could not be easily converted to vector format due to their complexity or source limitations. We have done our best to enhance the clarity of these figures while retaining their original content.
- Regarding Consistency in Writing Physical Units
You noted inconsistencies in the writing of physical units, such as "Mpa" instead of "MPa." We have carefully reviewed the entire manuscript and corrected all similar typographical errors to ensure consistency and accuracy in unit notation.
8.Regarding Trend Lines and Regression Equations in Graphs
You recommended incorporating trend lines, regression equations, and determination coefficients (R²) into the graphs to improve data readability and scientific rigor. We have followed this advice and added fitting curves for the experimental data in Figure 14. These curves highlight the data trends and provide regression equations along with R² values, aiding readers in understanding the underlying relationships. For instance, Figure 14 includes a linear regression equation with an R² value that quantifies the goodness of fit. By including these elements, we aim to clarify the relationships between variables and offer readers valuable insights for further analysis and prediction.
9.In the manuscript, the information of the third corresponding author has been removed, retaining only the details of the second corresponding author, as highlighted in red in the author information section. Additionally, the project number for the Key Research & Program of Zhejiang Province, China, has been updated from the incorrect number in the first draft to the correct one (Grant No. 2020B02011).
10.We have revised the units in Tables 1 and 2 (as highlighted in red in the tables) and provided explanations for the symbols of the relevant physical quantities.
We are very grateful for your review comments and believe that these revisions will significantly enhance the quality and scientific value of our manuscript. We have made comprehensive revisions according to your suggestions and look forward to your further feedback.
Thank you very much for your time and effort, and we await your re-review.
Best regards,
Jianqiang Zhang and Yanjun Qin
